# Bergamot Polyphenols Reduce Hepatic Lipogenesis While Boosting Autophagy and Short-Chain Fatty Acid Production in a Murine “Cafeteria” Model of MASLD

**DOI:** 10.3390/nu17233684

**Published:** 2025-11-25

**Authors:** Concetta Riillo, Maddalena Parafati, Francesco Crupi, Bartosz Fotschki, Monica Ragusa, Anna Di Vito, Chiara Mignogna, Vincenzo Mollace, Elzbieta Janda

**Affiliations:** 1Department of Health Sciences, University Magna Graecia, 88100 Catanzaro, Italy; concettariillo@unicz.it (C.R.); francesco.crupi@unicz.it (F.C.); ragusamonica29@gmail.com (M.R.); mollace@unicz.it (V.M.); 2Department of Pharmacodynamics, College of Pharmacy, University of Florida, Gainesville, FL 32610, USA; mparafati@cop.ufl.edu; 3Institute of Animal Reproduction and Food Research, Polish Academy of Sciences, 10-748 Olsztyn, Poland; b.fotschki@pan.olsztyn.pl; 4Department of Veterinary Prevention, Provincial Health Authority, 98121 Messina, Italy; 5Department of Experimental and Clinical Medicine, University Magna Graecia, 88100 Catanzaro, Italy; divito@unicz.it (A.D.V.); mignogna@unicz.it (C.M.); 6Institute of Research for Food Safety & Health IRC-FSH, University Magna Graecia, 88100 Catanzaro, Italy

**Keywords:** cafeteria diet, autophagy, polyphenols, *Citrus bergamia*, steatosis, butyrate

## Abstract

Background: Non-alcoholic fatty liver disease (NAFLD) is the most common liver disorder in Western countries, characterized by excessive fat storage in the form of lipid droplets (LDs) in hepatocytes; it is also called Metabolic-Associated Steatotic Liver Disease (MASLD), if coexisting with at least one cardiometabolic risk factor. Bergamot polyphenols (BPF) have been shown to counteract hepatic LD accumulation through potent lipogenesis suppression and associated metabolic benefits in Wistar rats, but their liver-specific anti-lipogenic effects may be species- and strain-dependent. Results and Methods: To address this issue we examined the effect of a cafeteria diet (CAF) and BPF in C3H/HeOuJ mice, which are considered resistant to diet-induced MASLD and fibrosis. Interestingly, a 15-week CAF diet led to a robust LD accumulation, weak portal and focal inflammation, and induced a higher expression of lipogenesis (*Acly*)- but not fibrosis-related (*Col1a)* genes in C3H/HeOuJ livers. This was associated with a significant increase in blood glucose, triglycerides, and total cholesterol levels, and a decrease in caecal short-chain fatty acids (SCFAs). Importantly, mice co-treated with BPF showed strongly reduced steatosis compared to CAF mice, lower blood glucose and triglyceride levels, stimulation of hepatic autophagy, and a reduced *Acly* gene and protein expression, followed by a more than doubled and tripled production of total SCFAs and butyric acid, respectively. Conclusions: In conclusion, while CAF diet supplementation in C3H/HeOuJ mice proves to be a suitable model of MASLD with deficient SCFA production, BPF confirms its potency to prevent murine MASLD by pleiotropic mechanisms, including beneficial effects on SCFA content, hepatic autophagy, and lipogenesis.

## 1. Introduction

Non-alcoholic fatty liver disease (NAFLD) is defined as the pathological accumulation of LDs in >5% of hepatocytes in patients who do not consume excessive amounts of alcohol [1]. The more advanced stage of non-alcoholic steatohepatitis (NASH) is characterized by portal and lobular inflammation, hepatocyte ballooning, and fibrosis, which may progress to end-stage cirrhosis and hepatocellular carcinoma [1,2]. Patients with NAFLD or NASH are at a greater risk for cardiovascular diseases, as they typically present with several metabolic disorders such as obesity, insulin resistance, type 2 diabetes, hypertension, dyslipidemia, and other features of metabolic syndrome (MetS). For this reason, both the terms NAFLD and NASH are being gradually substituted with the new term “Metabolic Dysfunction-Associated Steatotic Liver Disease” (MASLD) that encompasses all liver diseases associated with pathologic fat accumulation and at least one cardiometabolic risk factor [3]. The current global prevalence of MASLD has been estimated to be 38% in adults, and it is increasing at an alarming rate in adolescents, reaching global epidemic proportions, representing a major social, economic, and public health issue [4,5]. Excessive caloric intake, high sugar consumption, limited physical activity, and genetic background represent the paramount risk factors for this condition [6].

Currently, there is no pharmacological treatment for MASLD, and lifestyle interventions, such as a healthy diet and regular physical exercise, remain the mainstay of management for patients with liver steatosis [7,8]. In this setting, nutraceuticals emerge as a promising avenue not only for prevention but also for the treatment of MASLD [9,10]. One of the candidates is BPF, which is an exsiccated bergamot fruit polyphenolic fraction. Bergamot (*Citrus bergamia*) is a *Citrus* family plant native to Southern Italy, characterized by its high content of unique flavanones in its fruits and leaves, which have garnered significant attention in recent years for their medicinal properties [11,12,13,14,15]. BPF’s chemical composition has been well characterized for scientific and commercial purposes [16,17]. It contains 58.2% flavanones, 21.6% flavones, 4.5% phenols, and 15.7% organic acids, limonoids, and other compounds, as quantified by peak areas. The most abundant flavonoids, namely naringin, neohesperidin, neoeriocitrin, bruteridin, and melitidin, account for 40% of the total BPF [16].

We have previously shown that the supplementation of BPF prevents NAFLD and NASH and facilitates the diet-based therapy of NASH in Wistar rats exposed to a CAF diet [10,18]. The term “cafeteria” (CAF) diet refers to a nutritional regimen based on a variety of energy-dense, sweet, and savoury snacks, commonly available in supermarkets and cafeterias, offered to laboratory animals ad libitum [13,19]. Unlike standard high-fat or high-sugar laboratory diets, the CAF regimen induces voluntary overconsumption, mimicking the same unhealthy eating patterns in humans and leading to rapid weight gain, fatty liver, and MetS in rodents [20]. These effects correlate with the stimulation of hepatic inflammation and inhibition of autophagy/lipophagy, and can be efficiently attenuated by the BPF treatment [18,21], leading to a potent suppression of lipogenesis, but not beta-oxidation gene expression programs in Wistar rat livers [13].

Although the liver-specific anti-steatotic action appeared as the most powerful effect of BPF in the Wistar/CAF model, BPF attenuated several other metabolic manifestations of MASLD, including increased triglycerides (TGLs), glucose and insulin levels in the blood, leptin resistance, glucose tolerance, inflammatory markers, and other effects. Similar effects of bergamot polyphenols, both as juice or leaf extracts, have been reported in other rodent models of MetS [14,22] as well as in clinical studies, where BPF exerted strong effects on hyperlipemia and hypercholesterolemia in patients with MetS [11]. Yet, it remains unclear whether the potent liver-specific anti-steatotic effects of BPF consistently occur alongside its systemic metabolic effects or are species- and strain-dependent.

In fact, in a murine model of NASH (DIAMOND mice on Western diet), no significant reduction in steatosis was detected; however, improvements in fibrosis and lobular inflammation were observed in BPF-treated animals, suggesting species-specific differences in response to BPF between mice and rats or a particular feature of this murine strain [23]. As a matter of fact, DIAMOND mice have been selected to develop NASH rapidly and are particularly prone to fibrosis [24]. To address this issue, we decided to perform a similar experiment on a different murine strain, C3H/HeOuJ (shortly C3H), which is considered resistant to NASH and fibrosis [25], but can develop mild hepatic steatosis [26]. To this aim, we applied the CAF diet, which might present several advantages over classical high-fat/sugar diets in MetS and MASLD induction [19,27] and assessed standard outcome measures of hepatic steatosis like total liver lipids and total LD area, alongside other parameters that reflect liver fat accumulation.

The second reason to use C3H mice is that they represent a good model organism for investigating the role of the gut microbiota and its metabolites, SCFAs, in MetS and MASLD pathogenesis [26,28]. In addition, these mice can be easily colonized with human microbiota to address the microbial interactions with specific diet components and their impact on SCFAs in more translational settings [29].

SCFAs are the end products of the fermentation of fibres and other dietary components by gut microbiota, which exert multiple beneficial effects on the gut physiology and many other host organs, including the liver, nervous, and immune systems [30,31,32]. They are absorbed from the distal intestine to participate in the regulation of energy metabolism and many metabolic functions of the host [31,33,34]. For example, acetate and propionate have been shown to reduce the hepatic gene and protein expression of lipogenic enzymes, thereby preventing liver steatosis [26,35], while butyrate attenuated the progression of MASLD by inhibiting Toll-like receptor 4 (*Tlr4*) mRNA and iNOS expression [36]. Importantly, dietary polyphenols have been shown to regulate the activity of intestinal microbiota and influence the production of SCFAs [9,37], but the impact of bergamot polyphenols on these compounds has never been investigated.

Here, we addressed the question of whether the CAF diet can boost MASLD development in C3H mice and how BPF supplementation influences this pathological process in the liver and in the gut. We show that C3H mice on the CAF diet for 15 weeks develop a strong steatosis with weak inflammation, and it can be well prevented by BPF. Bergamot polyphenols not only inhibit hepatic lipogenesis but also strongly stimulate hepatic autophagy, while inducing prebiotic effects on caecal SCFAs.

## 2. Materials and Methods

### 2.1. Animal Procedures and Experimental Design

Eight-week-old male (*n* = 24) C3H/HeOuJ mice, purchased from Charles River Lab. (Calco, Lecco, Italy), were housed three mice per cage in an animal housing facility and maintained in a temperature-controlled environment (20 ± 2 °C), on a standard 12 h light/dark cycle (7:00 a.m.–7:00 p.m.). Before allocation into one of the four experimental groups, the animals were provided free access to water and a standard chow (SC) diet 2016 (“SC”, Teklad, Harlan Laboratories s.r.l., Natisone (UD), Italy). All experimental procedures were conducted in compliance with ethical regulations and were approved by the Italian Ministry of Health and the local ethics committee (Authorization No. 758-PR-2019).

Nine-week-old mice were weighed, ear-marked for identification, and randomly allocated to one of two experimental groups: a standard chow (SC) diet group (SC, *n* = 12) or a CAF diet group (CAF, *n* = 12). After a week of adaptation to the new cage mate, the administration of the CAF diet started (day “0”) and lasted 15 weeks. The CAF diet composition and feeding protocol have been previously described [18], and the exact food products used in this study are listed in Appendix A. On the day 1, the CAF and SC groups were further divided into two subgroups: one was administered BPF extract (~50 mg/kg body weight/day) as a supplement in the drinking water (SC+BPF, *n* = 6; CAF+BPF, *n* = 6), while the other received drinking water without supplementation (SC, *n* = 6; CAF, *n* = 6). This dosage was extrapolated as the murine equivalent of the maximum daily human dose of 1000 mg (approximately 10 mg/kg), as previously described [15] and reported in several studies [18,38]. The daily dosage of 50 mg/kg/day was ensured by the procedure described in the Appendix A.

BPF^TM^, containing 80% of flavonoids purified by the size-exclusion separation of clarified juice components by a patented technique, was kindly provided by Herbal and Antioxidant Derivatives (H&AD) s.r.l. (Bianco (RC), Italy). The batch used in this study was characterized in detail for its chemical composition by Baron et al. [16].

Fresh BPF solutions and CAF diet items were provided every two days. Three mice of the same experimental group were caged together. The numerosity of 6 mice per group was calculated by G* power 3.1 software for the expected effect size of 0.6 for the total liver lipids in treated animals, according to our previous studies [10,13,18] and established based on the ethical principle of reduction. One animal was withdrawn from the study due to very low weight (below 17 g) at week 4.

Food consumption and body weight gain were monitored weekly. All animals were fasted for 4–5 h prior to euthanasia, which was performed under a Ketamine/Xylazine combination (80 mg/kg + 10 mg/kg). Blood was collected by cardiac puncture, and tissue collection was analysed as previously described [18]. Samples were collected, marked with codes, and stored at −80 °C until analysis. Two main investigators, C.R. and M.P., were aware of the coding and sample allocation. Pathological evaluations and data analyses were performed independently by researchers unaware of group identity. The final anaesthesia, surgery, and all the analyses were performed in sets of mice/samples representing each experimental group to reduce time differences between experimental groups and other confounding factors.

### 2.2. Blood Analysis and Total Liver Lipids Quantification

Haematochemical parameters were determined in the serum: total cholesterol, TGL, and glucose. The analyses were performed using commercial reagents on a Dimension EXL (Siemens Healthcare Diagnostics Srl, Milan, Italy). Total liver lipids were analysed according to the adapted procedure of Folch et al. [39], conducted as previously described [18].

### 2.3. Histology and Histochemistry of Mouse Liver Sections

Liver tissue was collected from the central region of the main lobe immediately after perfusion and processed using two separate staining protocols.

For haematoxylin and eosin (H&E) staining, a portion of the liver was fixed, dehydrated, and embedded in paraffin. Sections of 5 µm thickness were cut from each paraffin block and stained with Harris haematoxylin and eosin according to established procedures [18]. An expert pathologist (CM) evaluated these sections using the Histology Activity Index (HAI), which included scoring piecemeal necrosis (0–4), confluent necrosis (0–6), focal/spotty lytic necrosis (0–4), apoptosis and focal inflammation (0–4), portal inflammation (0–4), and Ishak fibrosis stage (0–6) [40]. All data were analysed using GraphPad Prism 9.0 (GraphPad Inc., San Diego, CA, USA).

For Oil Red O (ORO) staining, a separate portion of the liver, obtained from the same anatomical region but not embedded in paraffin, was snap-frozen and embedded in O.C.T. compound (Fisher Healthcare Tissue-Plus, Thermo Fischer Scientific Inc., Waltham, MA, USA) to preserve lipid content. Cryosections of 10 µm thickness were prepared and stained with ORO (Sigma-Aldrich, Merck Spa, Darmstadt, Germany), followed by haematoxylin counterstaining (ORO&H). Bright-field images were captured using a DM4000B microscope (Leica Microsystems GmbH, Wetzlar, Germany) and further analysed according to a procedure described previously [18].

### 2.4. Liver Sample Preparation and Western Blotting (WB)

Equivalent liver fragments (50 mg) were homogenized in a Glass Dounce Tissue Grinder (Wheaton, DWK Life Sciences GmbH, Wertheim/Mainz, Germany) at 4 °C in RIPA lysis buffer. Subsequently, both protein sample processing and WB were performed as previously described [18]. LC3 and ACLY blots were developed with ImmunoBlot ECL reagents (Cat. No. 170-5061; Bio-Rad Lab. Inc., Hercules, CA, USA), while ADRP blots were developed with highly cross-adsorbed goat (polyclonal) anti-rabbit IgG (H + L) antibody conjugated to IRDye 800CW 926-32211 and goat (polyclonal) anti-mouse IgG (H + L) antibody conjugated to IRDye 680RD (LI-COR Biosciences Inc., Lincoln, NE, USA). The following primary antibodies were purchased from Proteintech Group, Inc. (Rosemont, IL, USA): anti-LC3 (Cat. No. 14600-1-AP, 1:2000), anti-ACLY (Cat. No. 15421-1-AP, 1:2000), anti-α-Tubulin (Cat. No. 66031-1-Ig, 1:2000), and anti-ADRP/Perilipin 2 (Cat. No. 15294-1-AP; or from Santa Cruz Biotech. Inc. (Dallas, TX, USA): anti-β-Actin (Cat. No. sc-47778).

### 2.5. Quantitative (q)RT-PCR Analysis

For the detailed description of the qRT-PCR analysis and the sequences of primers used, see the Appendix A.

### 2.6. Analysis of Short-Chain Fatty Acids (SCFAs)

Caecal SCFA concentrations were measured by gas chromatography (Shimadzu GC-2010, Kyoto, Japan) according to a method described previously [41]. Briefly, samples were mixed with 0.2 mL formic acid, diluted with deionized water, and centrifuged at 7211× *g* for 10 min. The supernatant (1 μL) was injected onto a capillary column (SGE BP21, 30 m × 0.53 mm) using an on-column injector (Shimadzu, Kyoto, Japan). The oven temperature was ramped from 85 °C to 180 °C at 8 °C/min and held at 180 °C for 3 min. Injector and detector temperatures were 85 °C and 180 °C, respectively. Sample injections were performed in duplicate in two separate analyses. Calibration curves were prepared from standard mixtures of acetic, propionic, butyric, isobutyric, isovaleric, and valeric acids (Merck, Sigma, Poznan, Poland) and run every five samples to maintain accuracy.

### 2.7. Statistical Analysis

Statistical analyses were conducted using GraphPad Prism 10 (GraphPad Software, Inc.). Brown-Forsythe’s (BF) and Bartlett’s tests for equality of variances were performed, and if at least the BF test was negative, we assumed homoscedasticity. The majority of data were expressed as the mean ± standard deviation (SD) and evaluated using ordinary one-way ANOVA, followed by Tukey’s post-test or uncorrected Fisher’s LSD test. In some cases, two-way ANOVA was applied to evaluate data interaction (diet, supplement, diet × supplement). The mean fold change, in the parameters relative to CAF (XCAF) or CAF+BPF (XCAFBPF) group, was calculated according to the following formula: (XCAF ± SD)/XSC or (XCAFBPF ± SD)/XCAF, respectively, where XSC is the same mean X parameter for the SC group. For Ishak scoring, a Kruskal–Wallis test followed by Dunn’s multiple comparison test or uncorrected Dunn’s test was used. WB optical density (OD) was analysed as previously described [13,42]. The OD data are expressed as the mean ± standard error (SEM), since all technical replicates of WB samples were included in the analysis. Effect sizes for one-way and two-way ANOVA were calculated with eta-squared (η^2^) and partial eta-squared (η_p_^2^) formulas, respectively.

## 3. Results

### 3.1. Effects of BPF on Parameters Associated with CAF Diet-Induced MASLD in Mice

The experimental design of the study is illustrated in Figure 1. C3H mice were fed an SC diet until 9 weeks old, and then half of the animals were switched to the CAF diet for 15 weeks. Six animals from both the CAF+BPF and SC+BPF groups received daily BPF supplied with drinking water (50 mg/kg/day).

The CAF mice significantly gained body weight (BW) compared with the SC mice in the 11th week (Figure 2A), but no significant effect on BW was observed in the 15th week. The mean BW was not affected by BPF treatment in SC+BPF group, confirming previous studies [18]. Additionally, we observed an increase in the final BW in all groups, which was higher in CAF groups compared with the others (Figure 2B).

The CAF diet markedly elevated blood TGL levels by 2.4 ± 0.56-fold (Figure 3A), blood glucose by 1.7 ± 0.39-fold (Figure 3B), and total cholesterol levels by 1.46 ± 0.2-fold (Figure 3C), with respect to the SC group. A significant increase in liver weight was also observed in the CAF-fed mice compared with the SC group (Figure 4A), along with enhanced hepatic fat accumulation (2.1 ± 0.62) (Figure 4B). BPF (50 mg/kg per day) substantially attenuated these effects, restoring liver weight values to levels comparable with the control animals (Figure 4A). Although BPF did not significantly affect total cholesterol levels (Figure 3C), it considerably reduced TGL levels by 0.58 ± 0.13-fold (Figure 3A) and led to a modest reduction (0.8 ± 0.16) in blood glucose levels in CAF-fed mice (Figure 3B) compared to the CAF group, which is consistent with previous findings [10,18].

The analysis of liver samples revealed a significant reduction in total lipid content (0.73 ± 0.13) (Figure 4B) by BPF, which was confirmed by a strong suppression of the LD-coating protein ADRP/Perilipin 2 in CAF+BPF livers (0.56 ± 0.27) with respect to CAF samples (1.86 ± 0.6) (Figure 4C,D), but not in SC+BPF with respect to SC samples (Appendix A). Next, the analysis of ORO-treated liver sections from CAF-fed mice revealed an increase in the percentage area occupied by LDs (4.15 ± 1.2) when compared to the SC group. Importantly, BPF dramatically reduced this parameter in CAF+BPF sections by 0.47 ± 0.14-fold (Figure 4E,F). No significant differences were detected in any haematological, hepatic lipid, and LD parameters between the SC and SC+BPF groups. Overall, these findings indicate that BPF supplementation exerts a pronounced effect on TGL levels and hepatic lipid accumulation, a moderate influence on glycaemic control, and no observable impact on obesity-related parameters in CAF-fed C3H mice.

### 3.2. BPF Attenuates Hepatic Histopathological Alterations Characteristic of NAFLD and NASH in Murine CAF Model of MASLD

Next, we assessed whether BPF supplementation could ameliorate hepatic steatosis in CAF-fed mice. Livers from the SC and SC+BPF groups exhibited a uniform reddish coloration with minimal surface spotting (Figure 5A,B), and hepatocytes displayed consistent size and normal morphology (Figure 5C,D). In contrast, livers from CAF-fed mice appeared distinctly yellowish (Figure 5E,F) and showed extensive vacuolar degeneration in the midzonal regions and around central veins, indicative of severe steatosis (Figure 5G). Some hepatocytes in CAF-fed livers appeared swollen (white areas) and elongated, while perivascular regions showed pronounced inflammatory cell infiltration, reflecting inflammatory processes (Figure 5G). Remarkably, livers from CAF+BPF mice exhibited reduced vacuolar degeneration and a more organized hepatic structure compared to CAF controls (Figure 5H), suggesting that BPF treatment attenuated inflammation and liver injury. Histological analysis using the Ishak scoring system confirmed that CAF-fed mice showed portal and focal inflammation, as well as confluent necrosis, but no significant differences for piecemeal necrosis (Figure 6A–D), and no evident fibrosis (Ishak Stage), as reported in Figure 6E. On the other hand, CAF+BPF-fed mice showed a slight but not significant reduction in liver inflammation features.

To examine the impact of BPF on diet-induced hepatic lipid accumulation, serial liver sections from four dietary groups were analysed using ORO&H histochemistry. The staining revealed normal hepatic architecture in the SC control group (Figure 7A,B), with LDs localized primarily around the centrilobular regions near the veins. In contrast, livers from CAF-fed mice exhibited marked lipid accumulation, characterized by numerous LDs within hepatocytes in all areas of the section (Figure 7C,D).

This steatotic phenotype was substantially diminished in the CAF+BPF sections (Figure 7E,F), indicating that BPF supplementation mitigates the hepatic fat deposition associated with the CAF diet. These findings confirmed the data presented in Figure 4B–F.

### 3.3. BPF-Induced Effects on Autophagy and Lipid Metabolism in Murine CAF Model of MASLD

We initially examined the hepatic expression of autophagy-related markers across all experimental groups. Equivalent central regions of the main hepatic lobe were collected and subjected to WB analysis to assess LC3 (LC3-I and LC3-II) protein levels. Compared with SC controls, CAF-fed mice displayed a marked reduction in LC3-II levels (0.73 ± 0.16), whereas a significant increase (2.13 ± 0.64) was observed in the CAF+BPF group compared to the CAF group (Figure 8A,B). LC3-I expression showed a similar pattern. These results indicate a suppression of autophagic flux in CAF-fed mice and a restoration of autophagy following BPF treatment. Concerning the expression levels of lipid metabolism-related protein, we examined the effect of the CAF diet and BPF treatment on ACLY protein levels by WB analysis. In contrast to our earlier findings in rats [13], the CAF diet in this study increased hepatic ACLY protein levels (1.75 ± 0.26) when compared to SC-fed mice. Notably, BPF treatment significantly reduced ACLY expression (0.75 ± 0.18) when compared to the CAF group (Figure 8C,D).

### 3.4. Bergamot Polyphenols Reduce but Do Not Suppress Lipogenesis Genes in CAF Mice Liver

To examine the transcriptional alterations induced by the CAF diet and long-term exposure to BPF, the differential expression of selected candidate genes was quantified using qRT-PCR. Compared to the SC group, the CAF group showed a significant upregulation of lipogenic genes *Srebf1, Acaca*, and *Acly*, whereas *Ppara* expression remained unchanged when mRNA from 4/6 individual animals was analysed separately (Figure 9A–D). In addition, the CAF diet showed no significant effect on *Tlr4,* an inflammation-related gene, and *Col1a1,* a fibrosis-related gene, suggesting that CAF-fed C3H mice are not prone to develop consistent NASH and fibrosis (Figure 9E,F). Interestingly, BPF supplementation significantly reduced only *Acly* expression compared to the CAF group (Figure 9C), but not the other genes involved in lipid metabolism.

### 3.5. Effects of BPF on Caecal SCFAs Production in CAF-Fed Mice

The caecum is the main part of the digestive system, characterized by a high content of postbiotic metabolites. The SCFA concentration was examined in caecal digesta as described in the Materials and Methods section. Dietary factors and BPF differentially affected caecal SCFA production in this study. The total SCFAs were strongly reduced in the caecum of the CAF group (0.47 ± 0.18) as were the acetic, propionic, and butyric acid concentrations and the butyrate percentage, when compared to the SC group. Comparison between the CAF+BPF and CAF group showed an increase in total SCFA production by 2.41 ± 0.7 times to levels comparable with the SC group. This was observed for all main acids analysed: acetic, propionic, and butyric, as well as for total SCFAs and the percentage of butyric acid. The putrefactive SCFAs (iso-butyric, iso-valeric, and valeric acids) were not significantly affected by the CAF diet and BPF (Table 1).

## 4. Discussion and Conclusions

The present study was designed to determine the chronic effect of BPF on hepatic steatosis and gut-derived SCFAs in CAF diet-induced MASLD in the C3H mouse strain, by evaluating histological parameters and the accumulated fat in the liver, hepatic autophagy, inflammation, and lipogenesis gene expression. This murine strain has been used successfully to address the role of SCFAs in diet-induced MetS and hepatic steatosis, and in human microbiota colonization experiments, but has been recently classified as a poor model of NAFLD, resistant to both lipid accumulation and NASH upon Western diet treatment [25]. Since the efficacy of the CAF diet to induce NAFLD and obesity has never been tested on C3H mice, this experimental approach would give us the opportunity to verify the suitability of this diet to better model NAFLD in the C3H strain, along with testing BPF effects. All in all, this experimental design allowed us to demonstrate for the first time that (1) the CAF diet efficiently induces NAFLD, and initial stages of NASH, but no evident fibrosis in C3H mice at 15 weeks of the treatment; (2) BPF treatment significantly attenuated the anti-steatotic and hypolipemic effects of CAF diet and had less significant impact on other MetS parameters in these mice; (3) BPF strongly influences the amount and type of SCFA produced in the gut; and (4) BPF-induced expression of the autophagy marker LC3-II was much stronger in this murine model than in Wistar rats [13,18].

The novel findings of this study, as listed in points 1 to 4, will be discussed below in more detail.

Point (1) According to our observations, feeding C3H mice for 15 weeks with the CAF diet leads to a strong increase in hepatic steatosis associated with mild portal and focal inflammation, but not with fibrosis. Importantly, in this strain, liver steatosis induced by the CAF diet is markedly more severe than that induced by the Western diet [25]. This type of diet contains similar amounts of energy and macronutrients (39% fat and 40% carbohydrates with 12% fructose and 2% cholesterol) when compared to the CAF diet, but is provided in the form of food pellets, and thus should not be confused with the CAF diet, which favours overeating [20]. In fact, the Western diet did not induce a statistically significant increase in steatosis in C3H mice, even though the authors used a more advanced technique for hepatic fat quantification and a twice-as-long Western diet treatment (32 weeks) compared to the CAF intervention in our study. In contrast, the same study demonstrated a more than six-fold higher fat accumulation in C57BL/6J mice fed the Western diet than in the control group [25]. Thus, the steatotic effect of the CAF diet in C3H mice is more pronounced than in the Western diet model, but less severe than that observed in the Wistar rats/CAF diet model [13,18]. Accordingly, the average increase in total lipids in the CAF group compared to the control SC group (CAF/SC ratio) quantified by Folch’s method is 2.1 times in C3H mice and 4.4 times in Wistar rats. This corresponds to an increase in the total LD area, calculated as the CAF/SC ratio, by 4.2 and 10.4 times in C3H and Wistar animals, respectively.

Our results suggest that the CAF diet is a much more efficient strategy to induce hepatic steatosis than the Western diet and can be successfully applied to a “NASH-resistant murine strain” such as C3H mice. It is also possible that upon longer CAF diet treatment, these mice could develop fibrosis or at least a robust inflammation, which in this study, at week 15, could be classified as mild and detectable only at the histological level with moderate infiltration of immune cells and occasional ballooning and necrosis, but not yet by significant *Tlr4* gene expression.

The reason why the CAF diet is more efficient than the Western diet in inducing NASH can be attributed to its palatability and orosensory properties that cause hedonic overconsumption by activating the gustatory cortices and stimulating the reward system [20]. This hyperphagic effect that mimics human over-eating is almost totally absent in other high-fat/high-sugar diets, such as the Western diet, leading to much lower energy intake, and as a result, less steatosis and associated metabolic alterations compared to the CAF diet [43,44,45]. However, other properties of the CAF diet, such as the presence of highly processed foods and associated chemical additives, may also strongly contribute to NASH pathogenesis [27].

Point (2): Potent MASLD-related effects of CAF diet can be efficiently attenuated by BPF in C3H mice. The reduction was particularly strong and highly significant for hepatic LD accumulation (−52.5% vs. CAF, −69.2% vs. CAF-induced change (inc.), effect size 0.75), LD marker, ADRP, expression (−43.1% vs. CAF, −93.2% vs. CAF inc., effect size 0.46) and hepatic total lipid content (−26.8% vs. CAF, −53.5% vs. CAF inc., effect size 0.62), but slightly weaker for blood TGLs (−19.6% vs. CAF, −46.5% vs. CAF inc., effect size 0.34). These parameters depend on the balance between hepatic lipid synthesis, beta- and gamma-oxidation, and lipid transport, suggesting that these liver pathways are the primary target of bergamot polyphenols. Other MetS parameters, such as obesity, blood glucose, and cholesterol levels, were less affected by BPF treatment, suggesting that they are indirect targets of BPF. This is in line with our observations in Wistar rats, where we showed a strong reduction in steatosis, and outstanding suppression of lipogenesis genes, but less impact on blood glucose, cholesterol, and body weight [13]. The final important finding of this study is that BPF suppresses the expression of lipogenesis genes also in the livers of C3H mice, but the data are statistically significant only for Acly, the first enzyme of the lipid synthesis pathway, both at mRNA and protein levels, while there is a tendency for the expected modulation of other lipogenesis-related genes.

Furthermore, these data suggest that the hepatic fat reduction seems to be the most striking and common effect of bergamot polyphenols in the majority of quantitatively analysed NAFLD models. In fact, similar effects of bergamot polyphenols on hepatic lipid content and several other MASLD parameters were also demonstrated in Wistar rats treated with a high-sugar/fat diet [46]. On the contrary, the lack of steatosis reduction but presence of anti-inflammatory effects in response to BPF treatment in DIAMOND mice might reflect a unique feature of this strain rather than a common rule. In fact, it is likely that this mouse strain, which is particularly prone to developing NASH and fibrosis, very rapidly achieves maximum levels of steatosis and progresses to inflammation when fed a Western diet, and the effects of BPF supplementation are significant only at the level of inflammatory markers but are negligible at the level of lipogenesis suppression. However, it should be emphasized that the issue still remains open, as the authors did not demonstrate conclusively their findings. In fact, the study lacks any quantitative evaluation of hepatic lipids and formulates its conclusions merely on histological grading of steatosis.

Point (3): The most striking effect of the CAF diet is a significant reduction in total SCFA production for all SCFA types, except for all putrefactive SCFAs (PSCFAs) iso-butyric, iso-valeric, and valeric acids. The reduction in SCFA content is found in the majority, but not in all, diet-induced MASLD rodent models, and to our knowledge, it has also been investigated in CAF diet-fed rats, but not yet in murine strains. The SCFA modulation by the CAF diet was variable in rats, ranging from a slight increase to a significant depletion [47,48,49]; however, in none of the cases did SCFAs decrease so profoundly and drop to almost 40% of control group SCFAs, as described here, suggesting that SCFA-producing microbiota of C3H mice are particularly sensitive to CAF diet. Importantly, the continuous BPF administration was fully sufficient to reverse the harmful effect of the CAF diet on total SCFA production, and on the three most abundant compounds, acetic, butyric, and propionic acids. The significant reversion of SCFA production by BPF is a particularly interesting finding and requires further investigation. Other polyphenol preparations have milder effects and only partially revert the effects of hypercaloric diets, while in some other cases, such as phenolic acids [50] or raspberry polyphenols, they even reduce SCFA levels, while still inducing beneficial effects on gut microbiota [41]. A plausible explanation for the BPF effect may be that bergamot polyphenols—rather than oligosaccharides, which are absent in the extract—are metabolized in the gut into primary metabolites that act as precursors for SCFA synthesis through secondary metabolic pathways [30,51].

Regardless of the source of substrates for SCFA production, it is the metabolic activity of gut bacteria that primarily drives the changes in SCFA levels found in the CAF and CAF+BPF groups, which should reflect the changes in the respective microbiota profiles. The CAF diet, in this respect, seems to be the most powerful diet to induce dysbiosis, characterized by a low alpha- and beta-diversity of bacterial strains, although it does not seem to be associated with a classical increase in the Firmicutes/Bacteroidetes ratio as in other MASLD models [27,48]. On the other hand, other polyphenols and their metabolites are known to modulate microbial populations by increasing their diversity and the growth of healthy bacteria, such as *Bifidobacterium*, *Lactobacillus*, and *Akkermansia* [37]. Our previous report showed that the treatment of Wistar rats with micronized bergamot fibres and BPF reverted the dysbiosis induced by the Western diet by promoting the growth of *Bacteroidetes* [52]. Yet, the prebiotic effects of BPF alone on the CAF diet in rodents remain to be investigated.

Nevertheless, the SCFA findings presented here suggest that the C3H/CAF murine model recapitulates very well observations in several human studies with respect to SCFA deficiency in obese subjects, and their increase upon polyphenol supplementation as described in recent reviews [9,37].

Point (4): Another important finding of this paper is a very clear regulation of hepatic autophagy in C3H mice associated with MASLD. We observed significantly reduced levels of LC3-II in CAF-fed livers with respect to control mice and nice autophagy stimulation in response to BPF treatment in the CAF, but not in the SC+BPF mice. Of course, this relatively easy detection of LC3-II modulation was facilitated by its high levels, i.e., higher LC3-II/LC3-I ratios in control C3H mice than in Wistar rat livers, in which we also reported a similar modulation of autophagy by BPF and CAF diet [13,18]. Another factor might be a better recognition of murine LC3 proteins with respect to rat counterparts, with the antibody used in the present study.

Autophagy is dysregulated in hepatic pathologies, and several studies have demonstrated a strong association between impaired or deficient autophagy and the development and progression of MASLD. Autophagy contributes to LD degradation by lipophagy, but as a homeostatic process responsible for macromolecular degradation, it also reduces stress responses associated with lipotoxicity, like unfolded protein response (UPR) and proteostasis [53]. Restoring autophagic function has been considered a key approach to mitigating hepatocellular injury [54,55] and BPF clearly boosts this process.

In conclusion, the data presented here suggest the use of BPF as a potential nutraceutical strategy to prevent MASLD that conjugates prebiotic effects on gut microbiota and metabolites, with potent suppression of hepatic lipogenesis and efficient stimulation of autophagy. Furthermore, our findings indicate that the C3H strain fed with the CAF diet is a suitable tool to study the impact of plant polyphenols on diet-induced hepatic steatosis and the complexity of underlying modulatory mechanisms.

## Figures and Tables

**Figure 1 nutrients-17-03684-f001:**
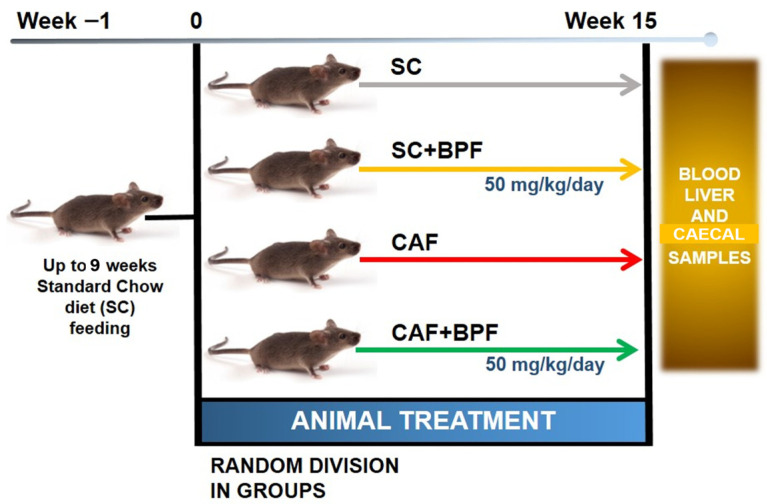
Experimental design of the study evaluating the effects of BPF on hepatic steatosis. A total of 24 C3H/HeOuJ mice were divided into groups (6 per group) based on dietary regimen and BPF administration (50 mg/kg/day) for a duration of 15 weeks. Before euthanasia, mice were fasted for 4–5 h, after which blood, liver, and caecal samples were collected for analysis.

**Figure 2 nutrients-17-03684-f002:**
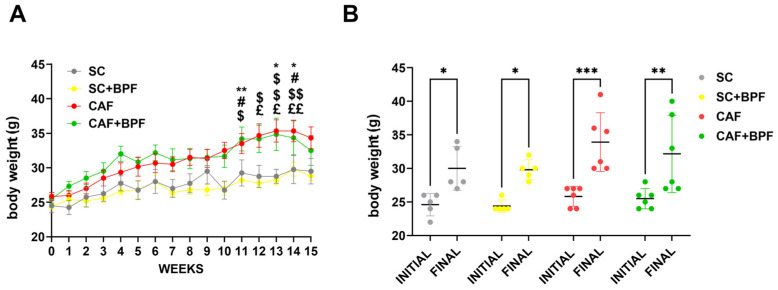
The CAF diet significantly increased BW, but BPF did not affect it. (**A**) Mean BW in four experimental groups over 15 weeks of the treatment. Data are presented as the mean ± SEM of *n* = 5/6 mice per group. (**B**) The initial and final BW. Data are presented as individual BWs and the mean ± SD (*n* = 5/6). Statistical analysis: Two-way ANOVA with Tukey’s multiple comparison test was used. (**A**) * *p* ≤ 0.05, ** *p* ≤ 0.01 SC vs. CAF; # *p* ≤ 0.05 SC vs. CAF+BPF; $ *p* ≤ 0.05, $$ *p* ≤ 0.01 SC+BPF vs. CAF; £ *p* ≤ 0.05, ££ *p* ≤ 0.01 SC+BPF vs. CAF+BPF. (**B**) * *p* ≤ 0.05, ** *p* ≤ 0.01, *** *p* ≤ 0.001.

**Figure 3 nutrients-17-03684-f003:**
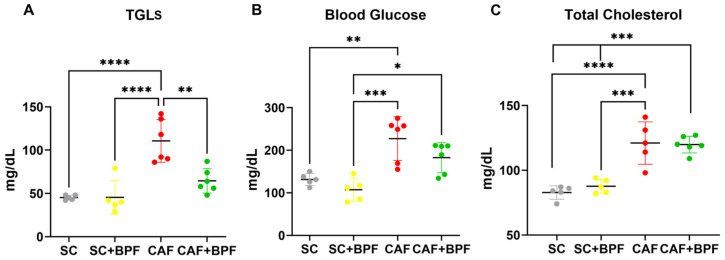
BPF induced a significant decrease in blood TGLs and a trend toward lower blood glucose levels in CAF diet-fed mice. Blood TGLs (**A**), glucose (**B**), and total cholesterol (**C**) were measured in cardiac blood-derived serum after 15 weeks of diet ± BPF treatment and 4 to 5 h starvation. Each dot represents a mouse. Data are presented as mean ± SD of *n* = 5/6 mice. Statistical analysis performed with two-way ANOVA with Tukey’s multiple comparison test, * *p* ≤ 0.05, ** *p* ≤ 0.01, *** *p* ≤ 0.001, **** *p* ≤ 0.0001.

**Figure 4 nutrients-17-03684-f004:**
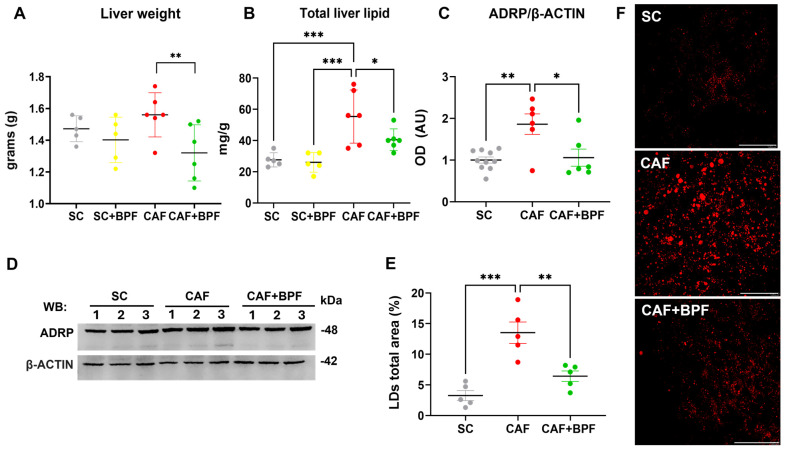
BPF intake decreases total liver mass and hepatic lipid accumulation in CAF diet-fed mice. (**A**) Final liver weights across experimental groups. (**B**) Total hepatic lipid content quantified by the Folch’s extraction method. Data are expressed as mean ± SD (*n* = 5/6 mice per group). (**C**) OD analysis of ADRP protein levels normalized to β-ACTIN; Mean ± SEM (*n* = 6/10 bands for each group). (**D**) Representative WB showing ADRP expression in three mice per group. (**E**) Graph showing the percentage of total area occupied by LDs; mean ± SEM (*n* = 5 mice per group, one dot = the mean of 5 images per mouse). (**F**) Representative confocal images of ORO-stained liver sections, transformed by threshold subtraction for LDs total area measurements. Scale bar = 40 μm. Statistical analyses: Two-way ANOVA with uncorrected Fisher’s LSD test (**A**,**B**) or ordinary one-way ANOVA with Tukey’s multiple comparison test (**D**,**E**). * *p* ≤ 0.05, ** *p* ≤ 0.01, *** *p* ≤ 0.001. AU (arbitrary units).

**Figure 5 nutrients-17-03684-f005:**
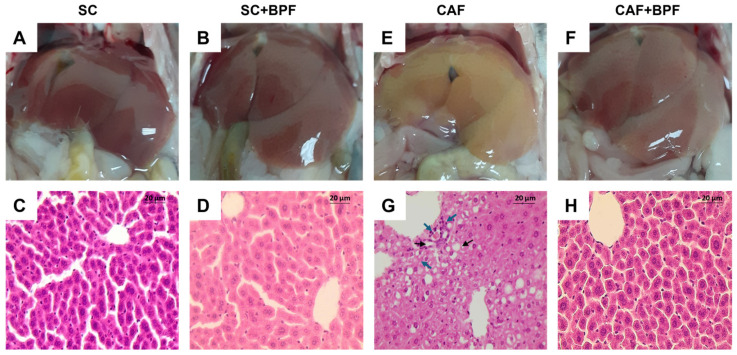
BPF attenuates histopathological markers of hepatic steatosis and inflammation in mice maintained on a CAF diet for 15 weeks. (**A**,**B**,**E**,**F**) Representative macroscopic images of livers captured following the abdominal cavity incision. (**C**,**D**,**G**,**H**). Representative liver sections stained with H&E. Scale bar = 20 µm; black arrows: apoptotic hepatocyte; blue arrows: inflammatory cell infiltration.

**Figure 6 nutrients-17-03684-f006:**
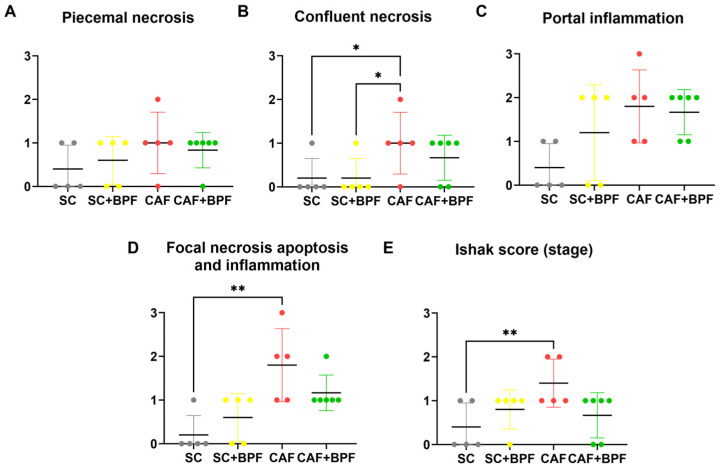
BPF reduces liver inflammation and necrosis in CAF-fed mice. HAI analysis was performed on liver sections to evaluate the following: piecemeal necrosis (**A**), confluent necrosis (**B**), portal inflammation (**C**), focal (spotty) lytic necrosis, apoptosis and focal inflammation (**D**), and Ishak fibrosis stage (**E**). Statistical analysis: Kruskal–Wallis test followed by uncorrected Dunn’s test (**B**,**C**) or Dunn’s multiple post-test (**D**,**E**). * *p* ≤ 0.05, ** *p* ≤ 0.01.

**Figure 7 nutrients-17-03684-f007:**
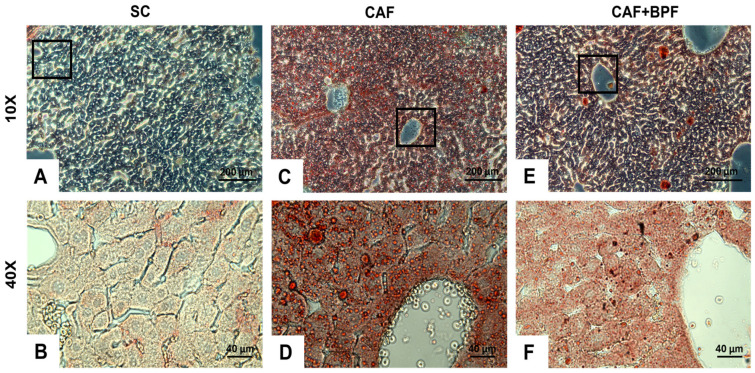
BPF attenuates hepatic lipid accumulation in CAF-fed mice. Representative liver sections from SC, CAF, and CAF+BPF groups were stained with ORO&H, highlighting neutral lipids and fatty acids in orange-red and cell nuclei in blue. Images were captured at 10× magnification (scale bar = 200 μm) for panels (**A**,**C**,**E**) and at 40× magnification (scale bar = 40 μm) for the corresponding higher-magnification views (**B**,**D**,**F**). The second-row images (**B**,**D**,**F**) show magnified regions corresponding to the black boxes in panels (**A**,**C**,**E**).

**Figure 8 nutrients-17-03684-f008:**
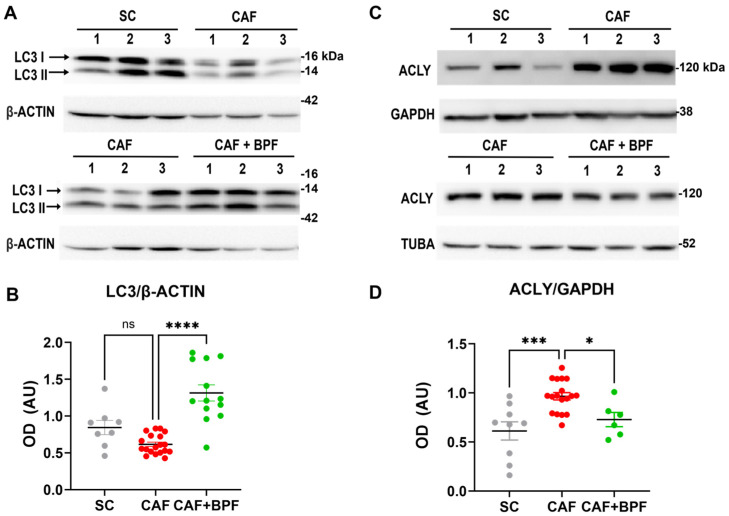
BPF restores autophagic flux and modulates lipid metabolism-related protein levels in the liver of CAF-fed mice. (**A**,**C**) WB images showing LC3-II and ACLY expression in liver samples from *n* = 3 representative mice per group. (**B**) OD quantification of LC3-II protein levels, normalized to β-ACTIN; mean ± SEM (*n* = 9/18 bands per group) (**D**) OD analysis of ACLY normalized to GAPDH; mean ± SEM (*n* = 6/18 bands per group). Statistical significance was determined using one-way ANOVA followed by Tukey’s post-test. * *p* ≤ 0.05, *** *p* ≤ 0.001, **** *p* ≤ 0.0001, ns = not significant. AU (arbitrary units).

**Figure 9 nutrients-17-03684-f009:**
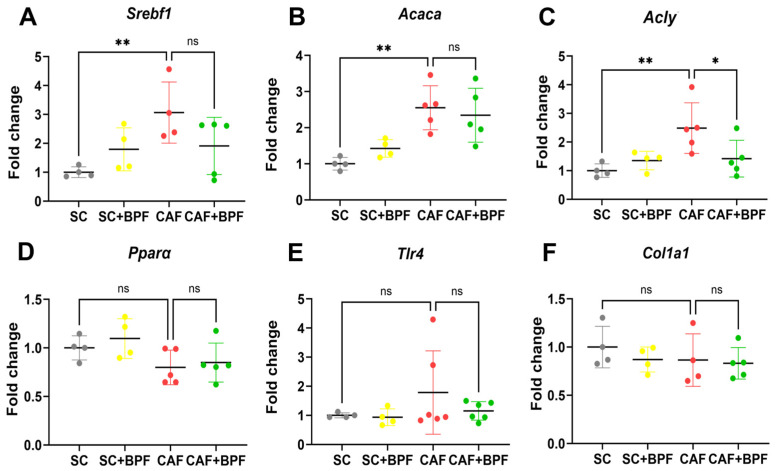
BPF reduces hepatic lipogenesis. (**A**–**F**) Gene expression levels were quantified by qRT-PCR and normalized to control (SC group) liver samples. Gene names are indicated above the figures. Each dot represents a mean value for one mouse. Data are expressed as mean ± SD from *n* = 4/6 mice. Outlier values were excluded. Statistical analysis: Two-way ANOVA followed by uncorrected Fisher’s LSD test. * *p* ≤ 0.05; ** *p* ≤ 0.01; ns, not significant.

**Table 1 nutrients-17-03684-t001:** The concentrations and percentage (% of total) of SCFAs compounds analysed in caecal digesta of four mouse groups. Mean ± SD of *n* = 5/6 mice. PSCFA, putrefactive SCFAs (the sum of isobutyric, isovaleric, and valeric acids). Values sharing the same superscript (^a^, ^b^, ^c^) within a row differ significantly as analysed by one-way ANOVA followed by Tukey’s post-test. The exact *p*-value for each pair is indicated in the last column. ns—not significant.

SCFAs Concentrations in Caecal Digesta											
	Group								ANOVA *p* Value		
	SC		SC+BPF		CAF		CAF+BPF				
		SDs		SDs		SDs		SDs	a	b	c
**SCFAs µmol/g digesta**											
**Total SCFA**	36.59	±8.48 ^a^	31.74	±5.89 ^b^	15.67	±11.12 ^abc^	37.77	±10.84 ^c^	0.005	0.033	0.004
**Acetic**	23.83	± 4.54 ^a^	19.62	±2.46	11.03	±7.37 ^ac^	24.37	±6.62 ^c^	0.004	ns	0.004
**Propionic**	5.98	±1.52	6.12	±1.57	2.89	±2.33 ^c^	6.97	±2.52 ^c^	ns	ns	0.016
**Butyric**	6.78	±3.71 ^a^	5.99	±2.47	1.75	±1.64 ^ac^	6.43	±2.70 ^c^	0.023	ns	0.048
**Iso-butyric**	0.34	±0.11	0.52	±0.18	0.58	±0.07	0.31	±0.16	ns	ns	ns
**Iso-valeric**	0.36	±0.16	0.60	±0.22 ^b^	0.32	±0.21	0.27	±0.16 ^b^	ns	0.048	ns
**Valeric**	0.46	±0.12	0.47	±0.12	0.37	±0.20	0.43	±0.13	ns	ns	ns
**PSCFA**	1.16	±0.39	1.59	±0.52	1.27	±0.48	1.01	±0.45	ns	ns	ns
**% of total SCFAs**											
**Acetic**	65.53	±3.96	62.56	±5.80 ^b^	73.97	±8.912	65.42	±6.93 ^b^	ns	0.035	ns
**Propionic**	16.54	±3.82	19.14	±2.36	17.01	±4.16	18.08	±4.01	ns	ns	ns
**Butyric**	17.93	±5.64 ^a^	18.31	±4.50 ^b^	9.02	±5.30 ^ab^	16.50	±4.63	0.031	0.023	ns

## Data Availability

The raw data supporting the figures and conclusions of this article will be made available by the authors on request.

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
