# Peer review of "Bergamot Polyphenols Reduce Hepatic Lipogenesis While Boosting Autophagy and Short-Chain Fatty Acid Production in a Murine “Cafeteria” Model of MASLD"

_nutrients, 2025, doi:10.3390/nu17233684_

Round 1
Reviewer 1 Report
Comments and Suggestions for Authors
With real interest and pleasure, I read the manuscript entitled ” Bergamot polyphenols reduce hepatic lipogenesis, while boosting autophagy and short-chain fatty acids production in a murine “cafeteria” model of MASLD.” (manuscript ID: nutrients-3977904), nicely written by experienced researchers, Riillo and colleagues, based on a solid study.
I have only minor/optional comments/suggestions:
- Please, explain in the introduction what exactly “cafeteria” diet is, so that the Reader would know from the beginning what it is all about, even if from outside of the field.
- Besides, please, explain the “CAF” abbreviation upon its first appearance in the introduction.
- Lines 141-142: “BPF, as earlier prepared and characterized for polyphenol content [15], was kindly provided by Herbal and Antioxidant Derivatives (H&AD) s.r.l. (Bianco, RC, Italy).”. Still, please, consider providing the Reader with the exact composition also in this manuscript, e.g. in the supplement.
- Were any similarities observed in liver expression patterns or metabolic changes compared to PMID: 30811073?, the study in which expression patterns have been analyzed in NAFLD models in an unbiased manner?
- Please, consider homogenization of the graphs throughout the manuscript, including the supplement. At the moment, they seem represent multiple options, actually each and every graphs seems to be different, including dot plots, bars, violin plots (the latter two with/without data points), with varying presentation of error bars, etc. It would be optimal if data points were present in the final, homogenized layout.
- Table 1 is fully technical/methodological and should go to the supplement.
- Table 2 seems to have two different titles. Besides, please, use “.” Not “,” for decimals.
- The latter applies also to Table S2 and, actually, everywhere where relevant.
- Lines 96-98: “In addition, these mice can also be easily colonized with human microbiota to address their interaction with diets and their impact on SCFA, obesity, MetS and MASLD [27].”. This aspect should be even more elaborated, i.e. in the discussion. Actually, polyphenols and SCFA precursors can interact/cooperate in some actions (PMID: 41135663).
- All abbreviations used in figures or tables should be explained in respective legends.
- Please, consider graphical abstract.
Author Response
EJ: We thank Reviewer 1 for the words of appreciation and the helpful feedback. Below there is point to point response to the specific comments:
- Please, explain in the introduction what exactly “cafeteria” diet is, so that the Reader would know from the beginning what it is all about, even if from outside of the field.
EJ: Thanks for your suggestion. In fact, “cafeteria” diet is not yet widely known in the field and as such it requires a larger introductory note. We added 3 explanatory sentences to the introduction of the revised version of our manuscript (Ms R1): Lines 68-73.
- Besides, please, explain the “CAF” abbreviation upon its first appearance in the introduction.
CAF abbreviation is defined in the abstract for the first time, but we have followed your suggestion to better explain the origins of the term “cafeteria” in the introduction.
Line 68. The term “cafeteria” (CAF) diet” refers to a nutritional regimen based on….
- Lines 141-142: “BPF, as earlier prepared and characterized for polyphenol content [15], was kindly provided by Herbal and Antioxidant Derivatives (H&AD) s.r.l. (Bianco, RC, Italy).”. Still, please, consider providing the Reader with the exact composition also in this manuscript, e.g. in the supplement.
EJ: We agree with the comment and accordingly, we added two sentences (Introduction, lines 61-65) that briefly characterize all main components of the extract. Importantly, the chemical composition of BPF extract was thoroughly analyzed in the papers by Formisano et al. 2019 and Baron et al. 2021 (cited in the text). In addition, besides the fact that inter batch variability is less than 7% (L. Tucci, personal communication), we used the same batch as one analyzed in Baron et al. For this reason, we made specific reference to this work in Materials and Methods (Line 144-145).
- Were any similarities observed in liver expression patterns or metabolic changes compared to PMID: 30811073?, the study in which expression patterns have been analyzed in NAFLD models in an unbiased manner?
EJ: We tried to compare the changes in Total lipid content and expression levels of MASLD-related genes with another model of rodent MASLD (Point 1 and 2 of Discussion) and we would like to consider another paper and dataset suggested by Reviewer 1. However, it is not easy to compare our data with the data presented in PMID: 30811073 for several reasons: 1) Total liver lipids or LDs are not analyzed in PMID: 30811073. This paper analyses genes expression levels in response to exercise in control and HFD mice. 2) the difference between CD (control diet) and HFD (high-fat diet) is not reported; in fact the expression levels in CD and HFD mice are normalized to 1 and the comparison is limited to “sedentary” and “after exercise” animals. 3) Finally, acute exercise (15 min and 3 h) does not induce significant changes in HFD-mice with respect to lipid metabolism genes.
- Please, consider homogenization of the graphs throughout the manuscript, including the supplement. At the moment, they seem represent multiple options, actually each and every graphs seems to be different, including dot plots, bars, violin plots (the latter two with/without data points), with varying presentation of error bars, etc. It would be optimal if data points were present in the final, homogenized layout.
EJ: Thank you for this helpful comment. The homogenization of the graphs was performed throughout the manuscript, and all the graphs were converted into dot plots, reporting all data points +/- SD or SEM where appropriate.
- Table 1 is fully technical/methodological and should go to the supplement.
EJ: Excellent suggestion. Table 1 was moved to Supplementary Information. We also moved the technical description of qRT-PCR methodology to keep the R1 Ms in 18 pages. Specific references were made in Material and Methods section, lines 133 and 203.
- Table 2 seems to have two different titles. Besides, please, use “.” Not “,” for decimals.
EJ: Thank you for pointing out this mistake. Table 2 now has one title. We also correct all 3 tables, one in the main Ms and two in Supplementary Info, as requested.
8.Lines 96-98: “In addition, these mice can also be easily colonized with human microbiota to address their interaction with diets and their impact on SCFA, obesity, MetS and MASLD [27].”. This aspect should be even more elaborated, i.e. in the discussion. Actually, polyphenols and SCFA precursors can interact/cooperate in some actions (PMID: 41135663).
8EJ: We thank the reviewer for this suggestion. Since the discussion is already very long we elaborated only on polyphenols as potential sources of SCFA synthesis (lines 498-501). We also cited the suggested literature the introduction in the context of anti-inflammatory properties of SCFA. (line 105).
9. All abbreviations used in figures or tables should be explained in respective legends.
EJ: The abbreviations introduced in the text such as CAF, TGLs, LDs, OD, WB etc.. are not explained. The abbreviations occurring only in the figures and tables were explained in the figure legends.
10. Please, consider a graphical abstract.
EJ: According to Reviewer’s suggestion, we prepared a graphical abstract that provides a summary of the main data presented in the paper.
Reviewer 2 Report
Comments and Suggestions for Authors
This study demonstrated that bergamot polyphenols (BPF) significantly reduce hepatic steatosis, suppress lipogenesis, and stimulate autophagy in C3H mice fed a cafeteria diet, a model for metabolic-associated steatotic liver disease (MASLD). BPF also restored short-chain fatty acids (SCFAs) production in the gut, suggesting potent prebiotic effects and highlighting its therapeutic potential for MASLD prevention. The paper was logically structured, the methodology was clear, and the results were well articulated. Here are my comments:
- Please ensure that all abbreviations are defined upon first use in the main text, such as line 60 “BPF”.
- Line 81, I recommend introducing the concept of 'species- and strain-dependent' effects earlier.
- Line 96 “SCFA” should be “SCFAs”.
- Line 133 “105-108”.
- Please replace the term line 144 "sacrifice" with a more specific and contemporary description of the procedure, such as "euthanized under anesthesia".
- Line 253, given that BPF was administered via drinking water, please clarify how the daily dosage of 50 mg/kg/mouse was ensured. Was water consumption monitored per cage or per mouse?
- A description of the data presented in Figure 4A is missing from the Results section.
- Please ensure that the panels in all figures are consistently ordered in the sequence they are cited in the text (e.g., A, B, C, D).
- To strengthen the mechanistic insight of this study, I recommend analyzing the gut microbiota composition in relation to the SCFA levels.
Author Response
Reviewer 2
This study demonstrated that bergamot polyphenols (BPF) significantly reduce hepatic steatosis, suppress lipogenesis, and stimulate autophagy in C3H mice fed a cafeteria diet, a model for metabolic-associated steatotic liver disease (MASLD). BPF also restored short-chain fatty acids (SCFAs) production in the gut, suggesting potent prebiotic effects and highlighting its therapeutic potential for MASLD prevention. The paper was logically structured, the methodology was clear, and the results were well articulated. Here are my comments:
EJ: We thank Reviewer 2 for his/her professional comments and positive feedback. Below there is point to point response to Reviewer’s specific comments:
- Please ensure that all abbreviations are defined upon first use in the main text, such as line 60 “BPF”.
EJ: We checked for the following abbreviations: TGLs, SCFAs, BPF, LDs, OD and WB.
2 . Line 81, I recommend introducing the concept of 'species- and strain-dependent' effects earlier.
EJ: Excellent suggestion. We introduced this concept already in the abstract of the revised R1 Ms, lines 19 and 20.
3. Line 96 “SCFA” should be “SCFAs”.
EJ: Thanks for pointing this out. In fact, short-chain fatty acids have been introduced as “SCFAs” in the abstract. We replaced “SCFA” with “SCFAs” throughout the R1 Ms.
4. Line 133 “105-108”.
EJ: We thank Reviewer for careful reading of the Ms. “105-108” was removed.
5. Please replace the term line 144 "sacrifice" with a more specific and contemporary description of the procedure, such as "euthanized under anesthesia".
EJ: Thanks for this suggestion. It was done in 3 cases.
6. Line 253, given that BPF was administered via drinking water, please clarify how the daily dosage of 50 mg/kg/mouse was ensured. Was water consumption monitored per cage or per mouse?
EJ: Thanks for pointing this out. As requested, we described the procedure to ensure the daily dosage of 50 mg/kg/mouse. Since the Materials and Methods section was already quite long, we moved this description to Supplementary Information, Supplementary Methods and specified it in the main R1 Ms (Line 139-141). The qRT-PCR method description was also moved to Supplementary Method to further shorten M&M section. This is the description of the procedure:
To ensure the daily dosage of 50 mg/kg/mouse the water consumption over 24 hours was monitored in mL per cage (Vd). To this end, 3 mice from the same cage were weighed to determine the total body mass of mice per cage (Tbm) expressed in kg and subsequently transferred to a metabolic cage for 24 h. To calculate the amount of BPF powder per cage per 2 days (Db) the following formula was used: Db = Vsuf x Dkg /Vd/Tbm, where Dkg is a daily dose per kg of animal mass and in this case 50 mg/kg and Vsuf is a sufficient amount of water (in mL) per 2 days, which should be at least 50% more than 2 x Vd. To prepare the beverage sufficient for 2 days, Db was diluted in Vsuf of drinking water. The beverages were refreshed every two days. To adjust the Db value to the increasing weight of mice, the beverage consumption measurement was repeated as above after 1 and 2 months from the study.
- A description of the data presented in Figure 4A is missing from the Results section.
EJ: It has been added to the revised version of the Ms in line 256.
8. Please ensure that the panels in all figures are consistently ordered in the sequence they are cited in the text (e.g., A, B, C, D).
EJ: Thanks for pointing this out this. We ordered the discussion of data according to the sequence of panels in Figures throughout the entire Results section.
9. To strengthen the mechanistic insight of this study, I recommend analyzing the gut microbiota composition in relation to the SCFA levels.
EJ: We thank reviewer for this excellent suggestion. Unfortunately, it is not possible to perform cecum microbiota analysis for this Ms as it was not planned in this experimental design, and the entire amount of cecum content was used up for SCFA analysis. We would have to repeat 15 week-long treatments of animals. Based on the data presented in this work we have planned to perform gut microbiota genetic analysis for future experiments on C3H mice.